# Genomes of a Novel Group of Phages That Use Alternative Genetic Code Found in Human Gut Viromes

**DOI:** 10.3390/ijms242015302

**Published:** 2023-10-18

**Authors:** Igor Babkin, Artem Tikunov, Vera Morozova, Andrey Matveev, Vitaliy V. Morozov, Nina Tikunova

**Affiliations:** 1Federal State Public Scientific Institution «Institute of Chemical Biology and Fundamental Medicine», Siberian Branch of the Russian Academy of Sciences, 630090 Novosibirsk, Russia; babkin@niboch.nsc.ru (I.B.); arttik@ngs.ru (A.T.); morozova@niboch.nsc.ru (V.M.); guterus@gmail.com (A.M.); doctor.morozov@mail.ru (V.V.M.); 2Department of Natural Sciences, Novosibirsk State University, 630090 Novosibirsk, Russia

**Keywords:** bacteriophage, caudoviricetes, genome sequence, stop codon recoding, phylogenetics, virus taxonomy

## Abstract

Metagenomics provides detection of phage genome sequences in various microbial communities. However, the use of alternative genetic codes by some phages precludes the correct analysis of their genomes. In this study, the unusual phage genome (phAss-1, 135,976 bp) was found after the de novo assembly of the human gut virome. Genome analysis revealed the presence of the TAG stop codons in 41 ORFs, including characteristic phage ORFs, and three genes of suppressor tRNA. Comparative analysis indicated that no phages with similar genomes were described. However, two phage genomes (BK046881_ctckW2 and BK025033_ct6IQ4) with substantial similarity to phAss-1 were extracted from the human gut metagenome data. These two complete genomes demonstrated 82.7% and 86.4% of nucleotide identity, respectively, similar genome synteny to phAss-1, the presence of suppressor tRNA genes and suppressor TAG stop codons in many characteristic phage ORFs. These data indicated that phAss-1, BK046881_ctckW2, and BK025033_ct6IQ4 are distinct species within the proposed *Phassvirus* genus. Moreover, a monophyletic group of divergent phage genomes containing the proposed *Phassvirus* genus was found among metagenome data. Several phage genomes from the group also contain ORFs with suppressor TAG stop codons, indicating the need to use various translation tables when depositing phage genomes in GenBank.

## 1. Introduction

Metagenomics is a powerful approach that significantly expands our knowledge about the world of microbes. In particular, the accumulated data on viromes from various habitats have provided an understanding that most of the virosphere is still “dark matter” [1,2,3]. The consistent application of the metagenomic approach to the study of this viral dark matter ensures the discovery and subsequent study of new groups of previously unknown viruses including viruses of bacteria. A number of new bacteriophages have been discovered in human gut viromes, namely, a highly abundant group of various CrAss-phages [4,5,6] and widespread megaphages [7]. The ubiquity of such phage genomes in the human viromes probably indicates the importance of such phages in the intestinal microbial community.

In addition to the abundant phage genomes, the genomes of rarer phages can be identified in gut viromes by bioinformatics methods. One of the reasons that complicates the analysis of phage genome sequencing data is the use of alternative genetic codes by some phages. It has been recently established that stop codon recoding is a relatively widespread phenomenon, occurring in the genomes of tailed phages from various families [6,7,8,9,10,11,12,13]. Suppressor tRNA genes have been found in the extended genomes of crAss-like, Black, Sapphire, Jade, and Agate phages [6,7,9,10,11,12,13]. Notably, some phage genomes with stop codon recoding do not contain the genes of suppressor tRNAs [6,7,8,12]. Previously, it has been suggested that the hosts of phages, which use stop codon recoding, also employ a similar translation strategy with alternative genetic codes [9]. However, the use of a standard genetic code has been shown for several hosts of such phages [6,8]. It has been assumed that stop codon recoding is a mechanism for regulating the expression of certain genes at the post-transcriptional level [8,9]. Thus, it has been detected that the presence of suppressed stop codons is characteristic mainly for the late phage genes, including the genes of structural proteins and proteins responsible for lysis of the host cell. In addition, the use of an alternative genetic code in a number of moderate phages can serve as a mechanism for switching the lysogenic pathway to lytic one [8].

In this study, the unusual complete phage genome was identified after the de novo assembly of the human gut virome. Most of the open reading frames (ORFs) encoding structural proteins and some other characteristic phage ORFs contained suppressor TAG stop codon/codons. Only two relative phage genomes were identified among the human gut metagenome data. Comparative genome analysis indicated that these three phage genomes substantially differ from those extracted from the available metagenomic data and genomes of the described phages.

## 2. Results

### 2.1. Analysis of the phAss-1 Bacteriophage Genome

Twenty virome shotgun libraries were constructed from fecal samples of healthy volunteers. In one of the obtained viromes (BioSample: SAMN37731570; Temporary SubmissionID: SUB13890116), the contig, with a length of 135,976 bp, that had an unusual but clear phage genome sequence was identified after sequencing and de novo assembly. An average coverage for the contig was 126 and it had a pseudo-circular structure, confirming that it is probably the complete genome of the unknown Caudoviricetes phage. It was named phAss-1 after “phage assembly”. The complete genome sequence of phAss-1 was deposited into the NCBI GenBank database with the accession number OQ749652. “Cut open” of a pseudo-circular scaffold was carried out in accordance with the result of the PhageTerm analysis.

After identification and annotation of the ORFs, several sequences encoding characteristic phage proteins, including terminase large subunit (TerL), tail tape measure protein, portal protein, and putative tail fiber protein, were found to be destroyed (Figure 1) due to the presence of TAG stop codons. When TAG stop codons in sequences encoding fragments of TerL, which is a conserved phage protein [14], were re-coded into glutamine residues (Gln), the complete TerL gene was composed. Notably, the re-coded gene showed clear similarity with some other phage TerL genes and suppressor Gln residues that occurred in positions that are conserved for non-supressor Gln positions in other TerL genes. Moreover, the phAss-1 genome contains three genes encoding suppressor tRNAs (tRNA-SUP-CTA), corresponding to the amber (TAG) stop codon. Analysis of the tRNA-SUP-CTA gene (Appendix A) using tRNAscan-SE v. 2.0 [15] indicated that this tRNA is responsible for Gln encoded by the amber stop codon. As the use of the suppressor TAG encoding Gln in phage genomes has been described previously [7,8,10], the translation table 15 (TAG -> Gln) was used to re-annotate the phAss-1 complete genome (Appendix A).

A total of 147 ORFs and 36 tRNA genes were identified in the phAss-1 genome. Of the 147 ORFs, putative products were predicted for 66 ORFs; the remaining 81 ORFs encoded hypothetical proteins. Clusters of genes encoding capsid and tail proteins, ORFs responsible for DNA replication and lysis were predicted in the phAss-1 genome. Forty-one ORFs (28%) were destroyed by suppressor TAG codons (Appendix A). Of them, 26 ORFs had assigned functions and most of the “recoded ORFs” are important for the phage life cycle. Notably, the ORF encoding specific aminoacyl-tRNA synthetase, which is necessary for the suppression of TAG stop codon, was not found in the phAss-1 genome. This is likely because this phage uses the host aminoacyl-tRNA synthetase.

### 2.2. phAss-1 Virome Comparative Analysis

Comparative ViPTree analysis of phAss-1 using the Virus-Host database indicated that phages with similar genomes were not described. However, BLASTN search using nt_viruses and nr/nt databases revealed two complete genomes with substantial similarity to phAss-1 (Table 1). The first sequence, BK046881 (*Caudoviricetes* sp. isolate ctckW2, 132,974 bp), was detected in the gut metagenome of an adult healthy man from the USA (BioSample: SAMN06164559 from BioProject PRJNA354235), the second one, BK025033 (*Caudoviricetes* sp. isolate ct6IQ4, 138,252 bp), was identified in the gut metagenome of a healthy infant from Denmark (BioSample: SAMEA2579955 from BioProject PRJEB6456). TAG suppressor stop codons were found in a number of genes from these phage genomes, including the characteristic phage genes (Figure 1). As such, both phage genomes were re-annotated using the translation table 15 (Appendix A). The phAss-1, BK046881_ctckW2, and BK025033_ct6IQ4 genomes and appropriate predicted proteins for alternatively coded phages are available through Zenodo (10.5281/zenodo.8324814).

The BK046881_ctckW2 and BK025033_ct6IQ4 genomes contain at least 142 and 143 ORFs, respectively, with predicted functions assigned to 57 and 64 ORFs. In addition, 28 and 32 tRNA genes were detected in the genomes, respectively, and both genomes carry two tRNA-SUP-CTA genes (Appendix A). Suppressor TAG stop codons were found in 37 and 41 ORFs in the BK046881_ctckW2 and BK025033_ct6IQ4 genomes, respectively, compared to 41 ORFs with amber stop codons in phAss-1. The phAss-1, BK046881_ctckW2, and BK025033_ct6IQ4 genomes demonstrate similar genome organization (Figure 1); they resemble blocks of ORFs encoding terminase subunits, structural proteins, enzymes of nucleic metabolism, and other metabolic enzymes, including ones responsible for host lysis. The phAss-1, BK046881_ctckW2, and BK025033_ct6IQ4 genomes contain long ORF encoding tail tape measure protein (3149, 3150, and 3209 aa, respectively) that is typical for phages with siphovirus morphology. Of the ORFs with predicted functions, these three genomes share 56 orthologous ORFs and their positions are conserved (Appendix A)

Despite the obvious similarity, there were certain differences in the phAss-1, BK046881_ctckW2, and BK025033_ct6IQ4 genomes. Eleven orthologous ORFs with assigned functions have various patterns of re-coding (Table 2). Thus, the ORF encoding ribonucleoside-triphosphate reductase activating protein contains suppressor TAG only in the BK046881_ctckW2 genome, whereas holin and dUTPase ORFs with suppressor TAG were detected in the phAss-1 and BK025033_ct6IQ4 genomes. Other ORFs with differences in the presence of TAG suppressor stop codons are given in Table 2. 

In addition, sequences of ORFs encoding tail tape measure protein, Ig domain containing protein, and putative flagellin-like protein of BK025033_ct6IQ4 substantially differ from those of phAss-1 and BK046881_ctckW2 (Figure 1). Moreover, 13 ORFs with predicted functions do not present in all three genomes (Table 2). A noticeable difference is in the receptor-binding tail fiber proteins of the phages. This protein (1070 aa) is encoded by one ORF in the BK046881_ctckW2 and BK025033_ct6IQ4 genomes, whereas two ORFs with the short intergenic spacer were found at the same locus in the phAss-1 genome (Figure 1). According to the NCBI protein Basic Local Alignment Search Tool (BLASTP; NCBI, Bethesda, MD, USA), the upstream short ORF encodes the structural tail protein (538 aa) and the downstream ORF—receptor-binding tail fiber protein (429 aa). 

Three-dimensional (3D) modeling of the receptor-binding tail fiber proteins that were encoded by the BK046881_ctckW2 and BK025033_ct6IQ4 genomes indicated the presence of four domains: two structural domains in the N-terminal part of the protein and two pectate lyase domains in the C-terminal part (Figure 2). Notably, two pectate lyase domains in this protein were determined despite low similarity. As for phAss-1, its structural tail protein (538 aa) contains the first two structural domains, similar to those of BK046881_ctckW2 and BK025033_ct6IQ4, whereas its putative receptor-binding tail fiber protein (429 aa) contains two pectate lyase domains (Figure 2 and Appendix A). Given the high similarity of structural domains encoded by the phAss-1 and two relative genomes, we can assume that the structural tail protein of phAss-1 is indeed the tail fiber protein connecting with the baseplate. Notably, two pectate lyase domains in the protein that is encoded by the downstream ORF of phAss-1 ORF were determined using despite low similarity sequence similarity with those of BK046881_ctckW2 and BK025033_ct6IQ4. It is unknown whether the phAss-1 protein with two pectate lyase domains connects with the receptor-binding tail fiber protein indicating different tail architecture of phAss-1 compared to that of two relative phages.

Nucleotide identity (NI) between phAss-1 and BK046881_ctckW2 or BK025033_ct6IQ4 genomes, which was calculated in the BioEdit 7.2.5 program [17], was 86.4% or 82.7%, respectively. NI between BK046881 and BK025033 was 80.4%. The calculated NI values are lower than 95% cut-off established by ICTV Bacterial Virus Subcommittee for the separation of a distinct phage species [18]. As such, taking into consideration NI values, similar genome synteny, the presence of suppressor tRNAs, and suppressor TAG stop codons in many characteristic phage ORFs, but the absence of specific aminoacyl-tRNA synthetase and certain differences in ORFs, we can suppose that phAss-1, BK046881_ctckW2, and BK025033_ct6IQ4 are distinct species within the unknown genus of the Caudoviricetes class. 

In addition, a group of phage sequences with low but definite similarity to phAss-1 and two relative phages were found using BLASTN search. Sequences with a length of less than 90,000 bp were excluded from the group; as such, 12 sequences were manually added to ViPTree analysis (Table 1 and Figure 3). Of them, nine and two sequences were extracted from human gut and oral metagenomes, respectively [19], whereas one sequence (MH552526_ctfi46) was found in the mouse metagenome. In addition, two genomes of phages with known host specificity, Enterococcus phage_EF1 (MF001358) and Enterocloster phage_PMBT10 (OQ326496), were added to ViPTree analysis (Figure 3). Importantly, of the added phage genomes, only three divergent genomes, namely, BK032759_ctxMM9, BK048656_ctOa11, and BK051641_ctUes22, have TAG suppressor stop codons in their ORFs. Of them, the tRNA-SUP-CTA genes were detected only in the BK048656_ctOa11 genome. In other genomes, including the genomes of Enterococcus phage_EF1 and Enterocloster phage_PMBT10, stop codon recoding was not found.

### 2.3. VIRIDIC Clustering

To clarify the taxonomy of phAss-1 and similar phages, a matrix of intergenomic similarities was constructed using the Virus Intergenomic Distance Calculator (VIRIDIC), described previously [20]. Sequences of phAss-1, 16 genomes from the Table 1, and ten genomes of phages from the Spbetavirus genus and Andrewesvirinae and Tybeckvirinae subfamilies (Figure 3) were used to construct the matrix (Figure 4). The values of pairwise similarity confirmed separating of phAss-1, BK035610_ctwa024, and BK040333_ct0TM7 into a particular putative genus that could be named the Phassvirus genus. In addition, relative sequences BK035610_ctwa024 and BK040333_ct0TM7 (NI = 88.1%), which were found in human oral metagenomes, as well as sequences BK015962_ctnPP24, BK035365_ct70u2, and BK037298_ctPL711 (NI varied from 68.3% and 74.0% between them) from human gut microbiomes, could form two new putative genera (Figure 4). 

Comparative genome alignment of the phAss-1 and 16 genomes from Table 1 (Figure 5) showed that synteny of the phAss-1, BK046881_ctckW2, and BK025033_ct6IQ4 genomes differed from the other 14 genomes. This indicated the separation of phAss-1, BK046881_ctckW2, and BK025033_ct6IQ4 into a distinct genus within this group of phages.

### 2.4. Phylogenetic Analysis of the phAss-1 Proteins

Phylogeny of several signature proteins, namely, TerL, major capsid protein, tail tape measure protein, and DNA polymerase III alpha subunit, was analyzed (Figure 6 and Appendix A). Of the proteins, TerL and tail tape measure protein contain suppressor TAG stop codons in the genomes of phAss-1, BK046881_ctckW2, and BK025033_ct6IQ4, and the corresponding ORFs in the genomes BK032759_ctxMM9, BK048656_ctOa11, and BK051641_ctUes22 were also re-coded using the translational table 15. Importantly, protein sequences of phAss-1, BK046881_ctckW2, and BK025033_ct6IQ4 phages formed a distinct well-supported branch in each phylogenetic tree (Figure 6 and Appendix A), indicating separation of the above phages into the proposed Phassvirus genus.

Corresponding protein sequences of phages, which are listed in Table 1, were grouped into a separate monophyletic cluster on each phylogenetic tree. In each case, the cluster contains a branch with the corresponding sequences of phAss-1, BK046881_ctckW2, and BK025033_ct6IQ4 phages (Figure 6 and Appendix A). Notably, corresponding sequences of phages ctwa024 and ct0TM7 from the oral microbiomes, sequences of phages ctnPP24, ct70u2, and ctPL711 from the human gut microbiomes that formed separate branches at the ViPTree (Figure 3), and separate groups on the matrix of intergenomic similarities (Figure 4) also clustered together on the constructed phylogenetic trees (Figure 6 and Appendix A). The obtained result indicated that phAss-1 and 14 extracted phage genome sequences comprise a new monophyletic group, which contains at least three putative genera, including the proposed Phassvirus genus, and could be a new putative subfamily or even family.

In addition, a phylogenetic tree was constructed for the unusual capsid protein sequences found in the phAss-1 and two other members of the proposed Phassvirus genus (Figure 7 and Appendix A). The genes encoding the proteins use suppressor TAG stop codons in the three above genomes.

Unexpectedly, the constructed tree contained only sequences of phAss-1, BK046881_ctckW2, and BK025033_ct6IQ4 phages (1665 aa, 1727 aa, and 1733aa, respectively). These three sequences formed a highly supported separate branch that grouped with the putative tail fiber protein (258 aa) of the Serratia phage 2050H1 (Ackermannviridae, Miltonvirus) that has myovirus morphology (Figure 7). Other phage genomes from the Table 1 do not contain ORF encoding similar capsid proteins, indicating that the presence of this unusual protein probably distinguishes members of the proposed Phassvirus genus form other phages from the group. Notably, the analyzed capsid protein of phages from the proposed Phassvirus genus seemingly has a multi-domain structure.

According to the analysis in the HHPRED program, N-terminal sequences with a length of approximately 900 aa show structural similarity with some phage head capsid proteins with high reliability. The most significant similarity (99% probability) was noted with the Enterobacteria phage RB49 highly immunogenic outer capsid protein (Hoc protein, phage capsid decorative protein). HHPRED additionally indicated significant similarity with the major capsid protein of the Bacillus phage phi29 (95% probability). Analysis using of protein structure homology-modeling SWISS-MODEL server showed the similarity of the above N-terminal sequences with that of Escherichia coli phage SU10 nozzile protein, which contains several immunoglobulin-like (Ig-like) domains [21]. InterProScan and BLASTP analyses confirmed that the Ig-like domain is located in the N-terminal part of the putative Hoc protein of phAss-1. 3D modeling of its structure using the AlphaFold2 program shows the presence of four Ig-like domains in its N-terminal sequence (Figure 8).

As for the C-terminal sequences (~800 aa) of the putative capsid protein of phAss-1 and two related phages, BLASTP analysis finds no similar proteins in the NCBI database. According to the analysis in the HHPRED program, the C-terminal sequences of this protein exhibit structural similarity with various prokaryotic proteins (probability 94–95%), namely, amino acyl peptidase of *Sporosarcina psychrophila*, methylamine dehydrogenase of *Paracoccus denitrificans*, and All3314 protein (putative contractile tail) encoded by the genome of *Nostoc* sp. strain PCC 7120. The AlphaFold2 program shows the presence of the beta-propeller domain in the C-terminal sequences of the putative capsid protein of phAss-1 and two related phages (Figure 8). Previously, beta-propeller domains have been found in the nozzle protein of *E. coli* phages T7 and SU10 [21,22]. Importantly, beta-propeller domains of phAss-1 and two related phages exhibit no detectable sequence similarity with those of T7 and SU10 phages. Notably, the nozzle proteins of T7 and SU10 contain a “platform” domain that is absent in the structural protein of phAss-1 and other Phassviruses. In addition, the beta-propeller structure was determined in the endosialidase of *E. coli* phage K1F, which is a T7-like phage. In K1F, the endosialidase is incorporated into its tail structure, whereas T7 contains the tail fiber proteins without such enzymatic activity [23].

### 2.5. In Silico Host Prediction

Since closely related phages for phAss-1 were not described, its possible that the bacterial host was assessed using nucleotide and amino acid sequence similarities. Sequence similarities between the phAss-1 genome and bacterial genomes were analyzed in the BLASTN program using the NCBI database. As a result, multiple matches of tRNA sequences of phAss-1 with bacterial tRNA sequences were detected. The highest similarity was found between the second tRNA-SUP-CTA of phAss-1 and that of *Dehalobacter* sp. (Bacillota; Clostridia; Eubacteriales; Desulfitobacteriaceae). In addition, the gene encoding DNA primase that shows high similarity with the orthologous phAss-1 gene was identified in the genome of *Dysosmobacter* sp. Marseille-Q4140 (Bacillota; Clostridia; Eubacteriales; Oscillospiraceae). Amino acid similarity of both proteins was 90%, identity—75% (LALIGN/PLALIGN software, access date 28 August 2023, https://fastademo.bioch.virginia.edu/fasta_www2/fasta_www.cgi?rm=lplalign). NI of the orthologous genes of DNA primase was 65.5% and related genes encoding amino acid sequences with an identity > 50% were not found in the NCBI database. Notably, no signs of prophage sequences were found in the region of DNA primase gene in the bacterial genome (FASTER software version 4.3, https://phaster.ca/). 

More evidence of the host range of phAss-1 was obtained by comparing all proteins encoded by the phAss-1 genome with the NCBI database of prokaryotic proteins using BLASTP. A search for similarities showed that most matches belonged to Bacteria; Bacillota; Clostridia; Eubacteriales; this may indicate that bacterial hosts of the phAss-1 phage could be a member of the Eubacteriales order of the Clostridia class.

## 3. Discussion

Bacteriophages, being very ancient creations, have a great variety in the organization of their genomes. Along with the utilization of modified nucleotides, the use of alternative genetic codes was found in some Caudoviricetes. In recent years, most of the data on phage genomes have been obtained from the results of metagenomic studies. The analysis of such genomes is an important part of reducing “dark matter” in the virosphere and comparative analysis is the first step.

Suppression of TGA and TAG stop codons in some phage genes can lead to their fragmentation into many short ORFs if standard genetic code is applied (translation table 11). However, only standard genetic code without alternatives can be used when depositing phages in the GenBank database. In that case, the correct annotation of phage genomes becomes impossible, and it is difficult to perform further comparative genome analysis. Without taking into account the use of stop codon re-coding by phages, it is hard to identify the existing diversity of phages and conduct a comprehensive analysis of viromes. As such, the use of various translation tables when depositing phage genomes in the Genbank database is required.

In this study, the unusual complete phage genome was found in the virome of a healthy person. Approximately 40% of all ORFs with predicted functions contained suppressor TAG stop codon/codons. Of the “damaged” ORFs, there were mostly virion genes and several genes encoding DNA-processing enzymes. In addition, three tRNA-SUP-CTA genes were identified in the genome, whereas the gene encoding specific aminoacyl-tRNA synthetase was not found. Importantly, signature phage proteins were encoded by ORFs with suppressor TAG stop codon/codons, indicating the importance of re-coding in the life strategy of this phage. The phage was named phAss-1 after the phage assembly.

A search for similar phage genomes revealed the absence of such genomes in the Virus-Host database. Only two similar genomes of Caudovorocetes phages (BK046881_ctckW2 and BK025033_ct6IQ4) were identified among human metagenomes. The NI level, similar synteny of the genomes, presence of signature phage ORFs with suppressor stop codons and tRNA-SUP-CTA genes, as well as certain differences between the genomes indicated that phAss-1 and these two genomes are different species within the same proposed *Phassvirus* genus. In addition, ViPTree, VIRIDIC, and phylogeny of the characteristic phage proteins showed that there is a monophyletic group of phage genomes identified in metagenomes (human gut, human oral, and mouse ones), and this group possibly forms a separate new family (Figure 3). *Enterococcus* phage_EF1 and *Enterocloster* phage_PMBT10 may also belong to this family. Further investigations are required to establish this potential family. Notably, some genomes from this monophyletic group also contain ORFs with suppressor stop codons. Within this group, two more possible genera are clearly distinguished, which, together with another genome, can form the potential subfamily. However, the taxonomic position of the putative *Phassvirus* genus within this monophyletic phage group remains unclear, since three members of this genus are distant from other phage genomes of this group (Figure 4). In addition, synteny of the *Phassvirus* genomes substantially differs from other genomes from the group and the unusual putative capsid protein was found only in the members of the proposed *Phassvirus* genus that confirms the separation of these three phages into the putative genus.

Only the complete or near-complete phage genomes were involved in the analysis in this study; however, a number of shorter sequences of phage genomes were found in the metagenome data, which also belong to this monophyletic but divergent group of phages (Figure 3). The corresponding sequences encoded by these particular genome fragments were contained in the cluster of the studied group of phages on each phylogenetic tree (Figure 6 and Appendix A). However, only three phage genomes from the proposed *Phassvirus* genus were extracted and even shorter related phage sequences were not found in the human, animal, and environmental microbiomes. Surprisingly, these three *Phassvirus* members (phAss-1, BK046881_ctckW2, and BK025033_ct6IQ4) were found in human gut samples collected at different times (2019, 2012, and 2008/2010) and in remote geographical locations (Russian Siberia, USA, and Denmark). It remains questionable which bacteria are the hosts of such rare phages (from the proposed *Phassvirus* genus). We could only assume that they are members of the Eubacteriales order of the Clostridia class. Taking into account that the tail architecture of phAss-1 differs from those of relative phages BK046881_ctckW2 and BK025033_ct6IQ4, we can hypothesize that phAss-1 has another host than two relative phages. 

An even bigger mystery is the role of these host bacteria in the human gut microbiota. It is known that the human intestinal microbiota is a dynamic system that contains obligate (core), transient, facultative (conditionally pathogenic), and even pathogenic microorganisms. The composition of the intestinal microbiota depends on many factors, namely, age, health, type of nutrition, individual habits (smoking, sports), and environment. Taking into consideration that the above samples were obtained from healthy people of different ages (two adults and an infant) that live remote places, we can only cautiously assume that the bacterial host/hosts for these phages could be transient bacteria and the bacteria is random for the human gut microbiome. 

As metagenomics studies provide a rapid accumulation of new data, some related genomes or genome fragments could be detected soon in addition to the three described rare phAssviruses. A detailed comparative analysis, starting with a correct deposit in the GenBank database, can help to clear up the taxonomy of the suggested *Phassvirus* genus and other members of the monophyletic group of divergent phage genomes noted in this study.

## 4. Materials and Methods

### 4.1. Virome Sequencing

Viral DNA isolation and sequencing from a fecal sample was described previously [24]. Briefly, the sample from a healthy volunteer was re-suspended in sterile phosphate-buffered saline and clarified by several consecutive centrifugations. DNase I, 5 U (Thermo Fisher Scientific, Waltham, MA, USA) was added to the final supernatant and the mixture was incubated for 4 h at 55 °C. Then, the mixture was treated with Proteinase K, 100 μg/mL (Thermo Fisher Scientific, MA, USA) supplemented with 20 mM EDTA, and 0.5% SDS for 3 h at 55 °C. Phenol-chloroform extraction with subsequent ethanol precipitation was used for DNA purification. The obtained DNA was diluted in 50 μL of 10% TE-buffer and, after measuring the concentration by Qubit 4.0 (Thermo Fisher Scientific, MA, USA), used in the standard procedure for constructing a virome shot-gun library using the NEB Next Ultra DNA library prep kit (New England Biolabs, Ipswich, MA, USA). A MiSeq Benchtop Sequencer (Illumina Inc., San Diego, CA, USA) and a MiSeq Reagent Kit 2 × 250 v.2 (Illumina Inc., CA, USA) were used for sequencing. The obtained sequences were assembled de novo using both the CLC Genomics Workbench software v.6.0 and SPAdes 3.15.

This work was approved by the Local Ethics Committee of the Center for Personalized Medicine, Novosibirsk (protocol #2, 12 February 2019), where this sample was obtained. Written consent of the healthy volunteer was obtained according to guidelines of the Helsinki Ethics Committee.

### 4.2. Genome Analysis

Phage termini of the obtained “pseudo-circular” scaffold were determined using PhageTerm [25] available on a Galaxy-based server (https://galaxy.pasteur.fr, access date 29 July 2023). For this, fastq dataset that was obtained after sequencing was used and linearization of the genome was carried out on the basis of PhageTerm analysis. To identify the putative ORFs and possible tRNA genes, the linearized genome sequence was imported into online RAST server v. 2.0 [26]. Functions of all identified ORFs were predicted by BLASTP (NCBI, Bethesda, MD, USA), InterProScan, and HPPred algorithms during the annotation of this genome that was named phAss-1. The prediction of the tRNA genes was additionally performed using the tRNAscan-SE V 2.0 server according to previously published recommendations [15]. Translation table 15 was used for re-annotation of the phAss-1 genome. In addition, manual verification of the annotation results was carried out using the NCBI GenBank protein database (https://www.ncbi.nlm.nih.gov, access date 19 July 2023).

Sequences of two most similar phage genomes BK046881_ctckW2 and BK025033_ct6IQ4 were also annotated for subsequent analysis based on the translation table 15.

### 4.3. Comparative Analysis

BLASTN was used for identification of nucleotide sequences similar to phAss-1 and BioEdit 7.2.5 program [17] was used for calculation of the intergenomic identity. Genome alignment was performed by the MAFFT program (https://mafft.cbrc.jp). Putative genera and species were predicted using thresholds of 70% and 95% of intergenomic identity, respectively [18,27].

Comparative analysis of phAss-1 sequence was performed using the ViPTree version 3.7 web server (https://www.genome.jp/viptree, accessed on 3 August 23) with default parameters [28]. ViPTree analysis was conducted based on genome-wide sequence similarities computed by tBLASTx and a viral proteomic tree was generated. The level of intergenomic similarity was determined using VIRIDIC (http://rhea.icbm.uni-oldenburg.de/VIRIDIC, accessed on 11 September 2023) according to [20]. 

### 4.4. Phylogenetic Analysis

To construct phylogenetic trees for phage proteins, amino acid sequences with significant similarity to the studied ones were extracted using the NCBI protein Basic Local Alignment Search Tool at the NCBI’s protein database. Protein sequences were aligned using the M-Coffee method in the T-Coffee program [29]. For all studied proteins, 50 most similar sequences were extracted, and then, incomplete sequences were removed. Maximum Likelihood phylogenetic trees were constructed in the IQ-tree program v. 2.0.6 [30]; the best fit substitution model according to ModelFinder [31] was used. Branch supports were assessed using 1000 ultrafast bootstrap replicates [32]. The resulting trees were midpoint rooted and visualized in FigTree v1.4.3.

### 4.5. 3D Modeling of Protein Structure

Ribbon and surface representations of proteins were predicted in the AlphaFold2 program available at https://colab.research.google.com/github/sokrypton/ColabFold/blob/main/AlphaFold2.ipynb, accessed on 11 September 2023 [33]. The resulting structural models were downloaded and edited using UCSF Chimera molecular visualizer, version 1.17 [34]. 3D protein models were constructed using a fully automated protein structure homology-modelling web server called SWISS-MODEL, available at https://swissmodel.expasy.org [35].

### 4.6. In silico Host Prediction

Possible bacterial hosts of phages were predicted based on the level of similarity of their genomes and sequences of encoded proteins with the NCBI database. Sequence similarities were analyzed in the BLASTN and BLASTP program using the NCBI nr database. Amino acid similarity of proteins was analyzed in the LALIGN/PLALIGN software, available at https://fastademo.bioch.virginia.edu/fasta_www2/fasta_www.cgi?rm=lplalign, accessed on 11 September 2023.

## Figures and Tables

**Figure 1 ijms-24-15302-f001:**
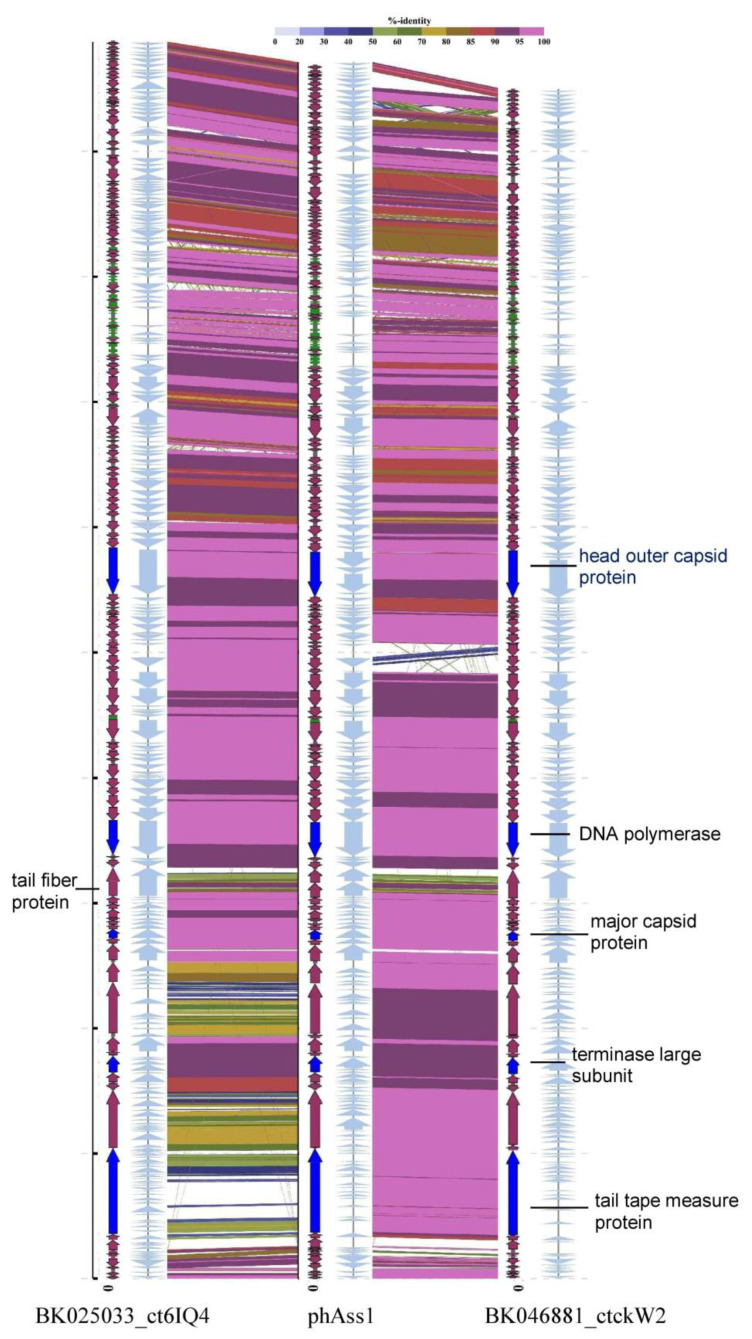
Comparative alignment of the phAss-1, BK046881_ctckW2, and BK025033_ct6IQ4 genomes. Colored lines between the aligned genomes indicate tBLASTx results. The color scale bar represents the percentage of identity. ORFs are represented by grey arrows (standard code) or colored arrows (TAG -> Gln recoding); the tRNA genes are highlighted in green; the ORFs encoding proteins that were additionally analyzed are colored in bright blue.

**Figure 2 ijms-24-15302-f002:**
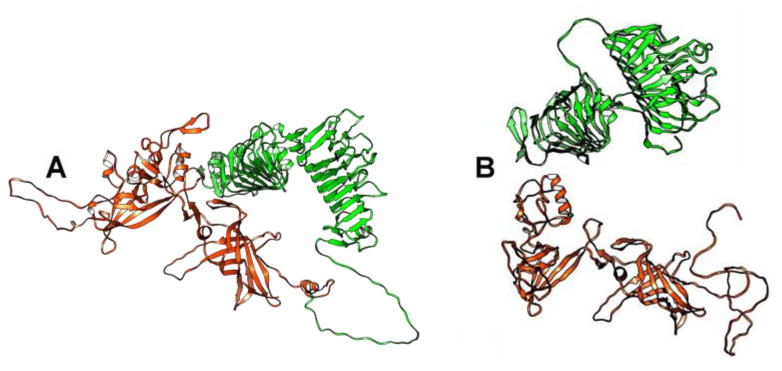
Ribbon representation of tail fiber proteins 3D structures encoded by the BK025033_ct6IQ4 and phAss-1 genomes. (**A**) Specified regions of the tail fiber protein of BK025033_ct6IQ4 are marked with red (1–475 aa, two structural domains located at the N-terminal part of the protein) and green (476–1069 aa, two pectate lyase domains located at the C-terminal part of the protein). (**B**) Structural tail proteins of phAss-1 that correspond to orthologous regions of the tail fiber protein of BK025033_ct6IQ4 are marked with the same colors—red for the tail fiber protein and green for the protein containing two pectate lyase domains. 3D models were predicted using AlphaFold2 and rendered using UCSF Chimera molecular visualizer, version 1.17.

**Figure 3 ijms-24-15302-f003:**
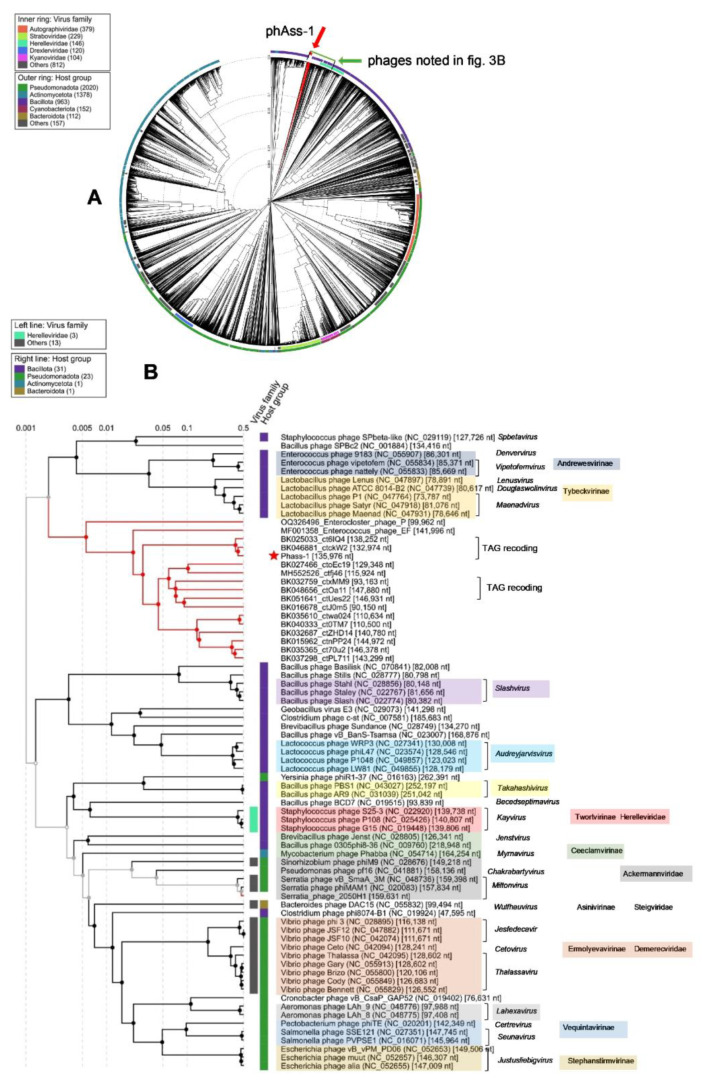
ViPTree analysis of the phAss-1 phage. (**A**) Circular dendrogram plotted by ViPTree using phAss-1 and phages with SG values > 0.001. The position of phAss-1 is marked with a red arrow. Phage sequences that were downloaded from the NCBI GenBank manually are marked with red phylogenetic branches. The bacteriophages shown in (**B**) are marked with a green rectangle. (**B**) The clade of the proteomic dendrogram indicating the position of the phAss-1 phage. The phAss-1 phage is marked with a red asterisk; phage sequences that were downloaded from the NCBI GenBank manually are marked with red phylogenetic branches. Phage genomes that have TAG suppressor stop codons in their ORFs are marked additionally.

**Figure 4 ijms-24-15302-f004:**
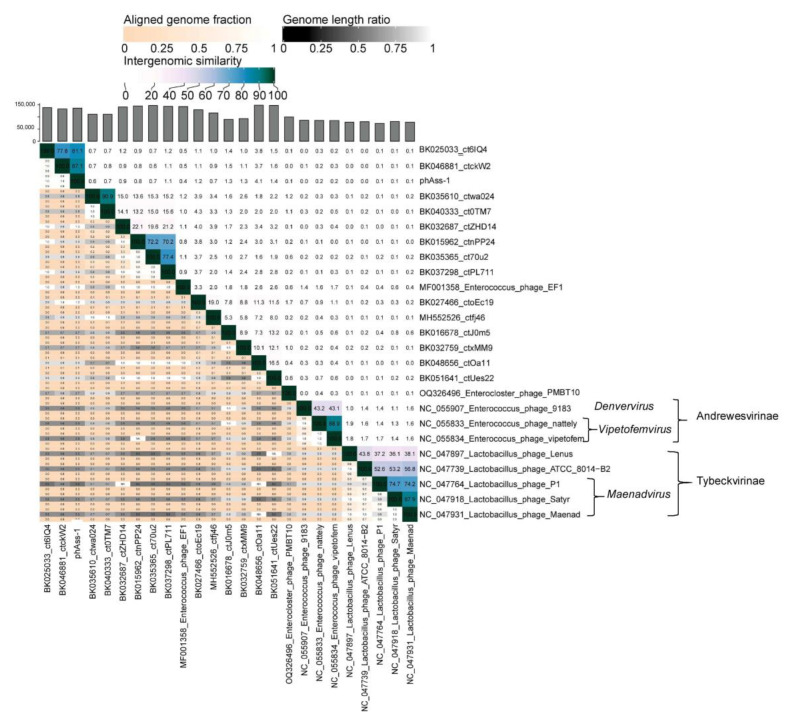
Bidirectional clustering heatmap visualizing VIRIDIC-generated similarity matrix for the phAass-1 and close phage genomes listed in Table 1.

**Figure 5 ijms-24-15302-f005:**
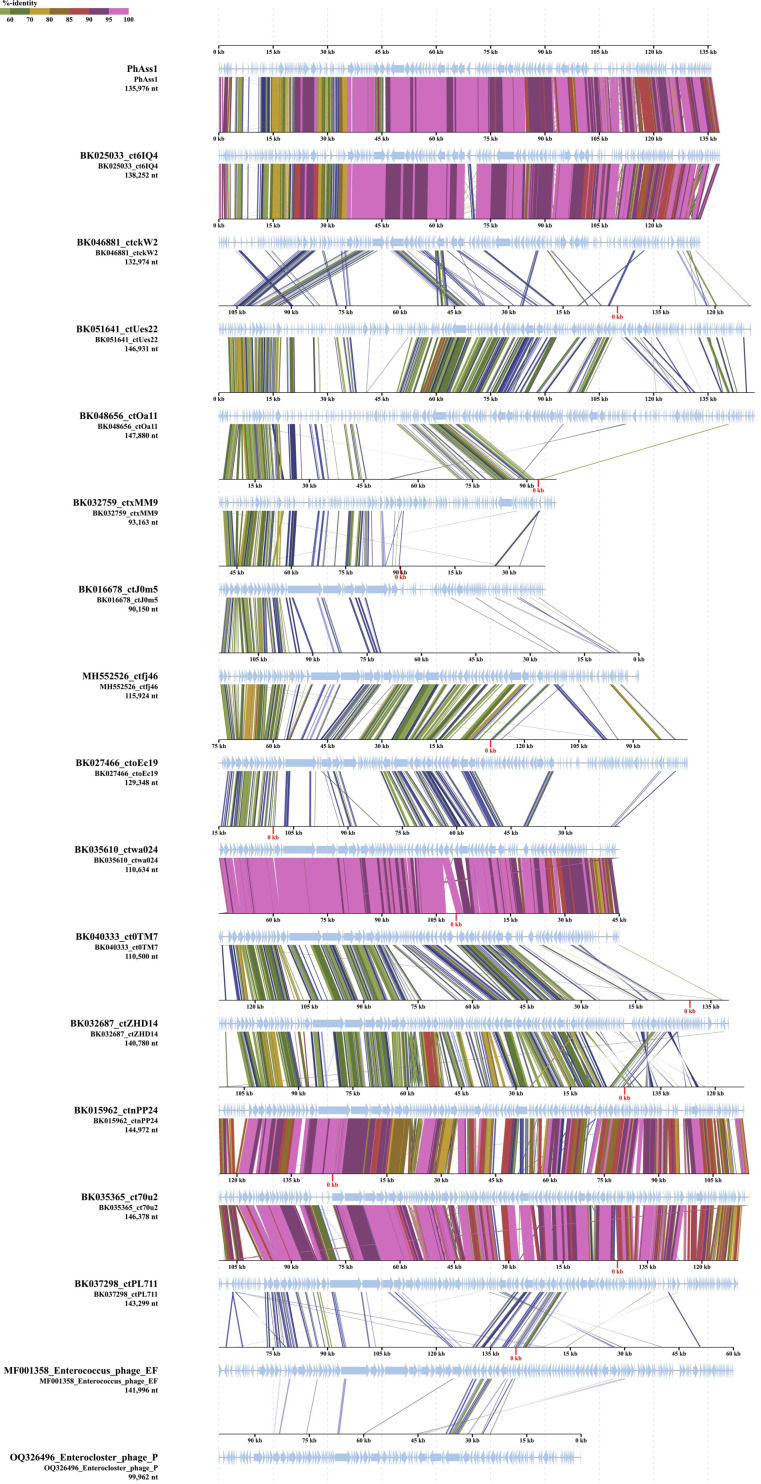
Comparative genome alignment of the phAass-1 and close phage genomes listed in Table 1. The percentage of sequence similarity is indicated in color; the color scale is shown at the top. Analysis was performed using ViPTree software version 3.7 web server (https://www.genome.jp/viptree, accessed 3 August 2023).

**Figure 6 ijms-24-15302-f006:**
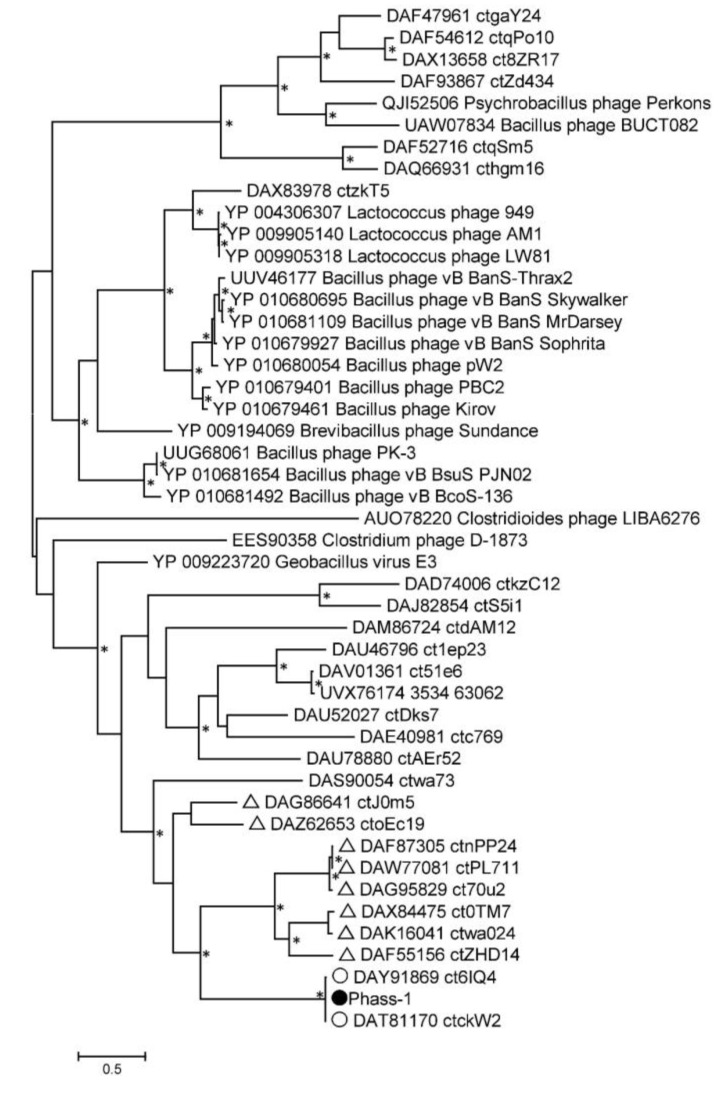
Maximum Likelihood phylogenetic tree of the phAss-1 terminase large subunit generated using IQ-tree software. The investigated sequence of phAss-1 is marked with a black circle; corresponding sequences of relative phages from the proposed Phassvirus genus are indicated with empty circles; sequences from phage genomes that are listed in Table are marked with empty triangles. Nodes with 95% statistical significance are marked with asterisks calculated from 1000 ultrafast bootstrap (UFBOOT) replicates. The scale bar represents the number of substitutions per site.

**Figure 7 ijms-24-15302-f007:**
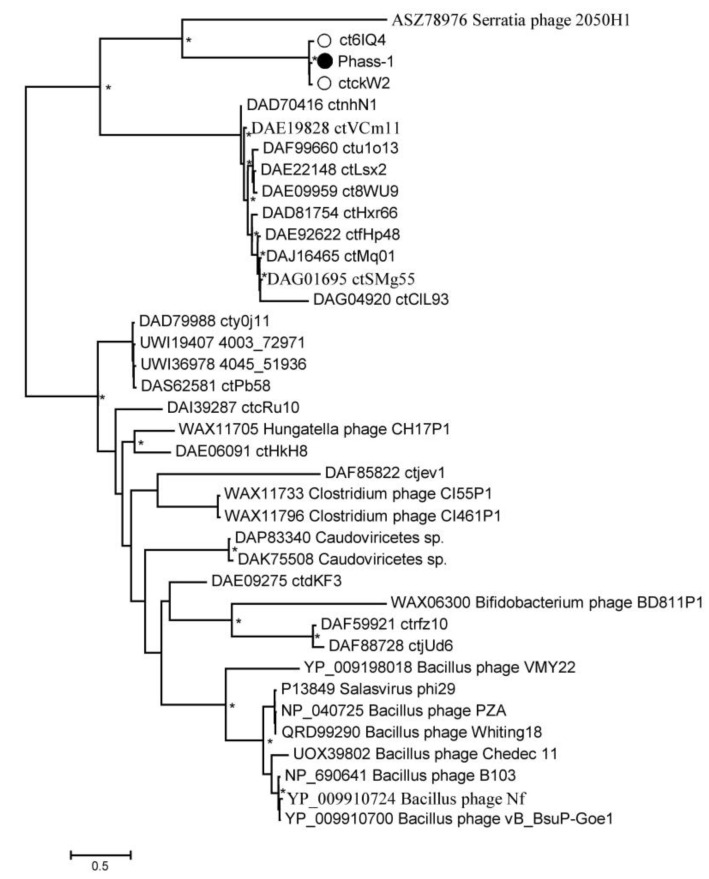
Maximum Likelihood phylogenetic tree of the phAss-1 structural capsid protein generated using IQ-tree software. The investigated sequence of phAss-1 is marked with a black circle; corresponding sequences of relative phages from the proposed Phassvirus genus are indicated with empty circles. Nodes with 95% statistical significance are marked with asterisks calculated from 1000 ultrafast bootstrap (UFBOOT) replicates. The scale bar represents the number of substitutions per site.

**Figure 8 ijms-24-15302-f008:**
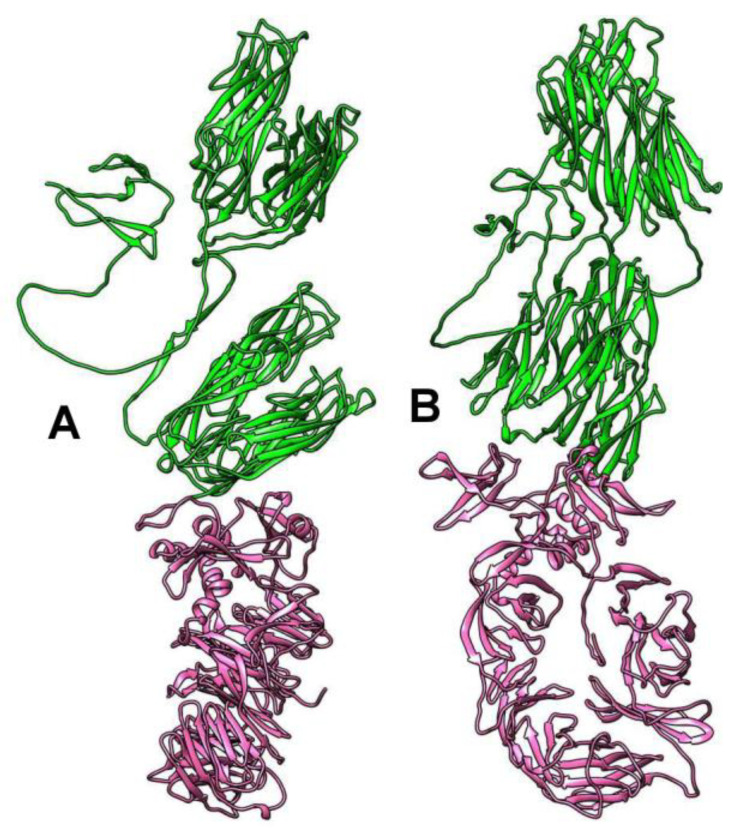
Ribbon representation of capsid protein 3D structure encoded by the phAss-1 genome shown in two orientations (**A**,**B**). Specified regions of the capsid protein are marked with green (1–900 aa, Ig-like domains located at the N-terminal part of the protein) and magenta (900–1666 aa, beta-propeller domain located at the C-terminal part of the protein). 3D models were predicted using AlphaFold2 and rendered using UCSF Chimera molecular visualizer, version 1.17.

**Table 1 ijms-24-15302-t001:** Phage genome lengths, sources, GC contents, and SG * values.

№	NCBI Accession Number and Name of Phage/Isolate	Genome Length	Source/Host	GC Content, %	SG	Mean Identity, %	Length, %
1	OQ749652_phAss-1	135,976	human gut metagenome	44.52	1	100	99.8
2	BK046881_ctckW2	132,974	human gut metagenome	44.39	0.8612	92.7	92.3
3	BK025033_ct6IQ4	138,252	human gut metagenome	44.45	0.8112	89.0	92.4
4	BK048656_ctOa11	147,880	human gut metagenome	35.33	0.0975	46.4	20.8
5	BK032759_ctxMM9	93,163	human gut metagenome	32.75	0.0741	43.4	11.2
6	BK051641_ctUes22	146,931	human gut metagenome	33.03	0.0651	43.1	14.8
7	MH552526_ctfj46	115,924	mouse tissue metagenome	30.75	0.0626	41.1	12
8	BK027466_ctoEc19	129,348	human gut metagenome	28.95	0.0596	42.6	12.4
9	BK016678_ctJ0m5	90,150	human gut metagenome	33.42	0.0528	42.3	7.8
10	BK040333_ct0TM7	110,500	human oral metagenome	35.22	0.0507	40.5	9.5
11	BK032687_ctZHD14	140,780	human gut metagenome	37.28	0.0502	42.8	11.5
12	BK037298_ctPL711	143,299	human gut metagenome	38.20	0.0499	41.0	11.9
13	BK035610_ctwa024	110,634	human oral metagenome	35.84	0.0496	40.8	9.4
14	BK015962_ctnPP24	144,972	human gut metagenome	38.72	0.0466	40.8	11.4
15	BK035365_ct70u2	146,378	human gut metagenome	38.36	0.0458	40.2	11.1
16	MF001358_Enterococcus_ phage_EF	141,996	*Enterococcus faecalis*	31.94	0.0281	39.1	6.9
17	OQ326496_Enterocloster_ phage_P	99,962	*Enterocloster bolteae* Q14	32.11	0.0123	39.7	2.1

* SG values were calculated according to Bhunchoth et al. [16] as normalized tBLASTx scores (SG; 0 ≤ SG ≤ 1) compared to phAss-1.

**Table 2 ijms-24-15302-t002:** Differences in ORFs with assigned function between the phAss, ctckW2, and ct6IQ4 genomes.

№	Function Assigned to the Specific ORF *	Phage Genomes		
		phAss-1	ctckW2	ct6IQ4
**TAG suppressor stop codons**				
1	ATP-dependent Clp protease	•		•
2	Ribonucleoside-triphosphate reductase activating protein		•	
3	Multiple antibiotic resistance protein MarR/DNA	•		
4	PhoH family ribonuclease		•	•
5	Phage holin	•		•
6	L-shaped tail fiber protein	•		
7	dUTPase	•		•
8	NADAR family protein	•		•
9	Antitermination protein, Q-dependent.	•		•
10	P-loop containing nucleoside triphosphate hydrolase	•	n.d. **	
11	Exoribonuclease		n.d.	•
**Presence in the genome**				
1	Receptor-binding tail fiber protein (1070 aa)		•	•
2	Structural tail protein (538 aa)	•		
3	Receptor-binding tail fiber protein/pectatliase (429 aa)	•		
4	LAGLIDADG endonuclease			•
5	Nicotinamidase	•	•	
6	Cysteine hydrolase			•
7	Transcription factor	•		
8	Nucleotidyltransferase-like protein	•		•
9	P-loop containing nucleoside triphosphate hydrolase	•		•
10	3′-5′ exonuclease	•		
11	Exoribonuclease	•		•
12	Phosphatase	•		•
13	RNAse	•		•

* ORFs encoding hypothetical proteins are not listed. ** n.d.—ORF was not found in the genome.

## Data Availability

Genomes and predicted proteins for alternatively coded phages are available through Zenodo (10.5281/zenodo.8324814).

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
