# Peer review of "Genomes of a Novel Group of Phages That Use Alternative Genetic Code Found in Human Gut Viromes"

_ijms, 2023, doi:10.3390/ijms242015302_

Round 1

Reviewer 1 Report

The authors report the identification of a novel phage from a single specimen. 

Major points:

1) a major limitation of this study is in the analysis of just one single sample. Analysis of a cohort of samples would greatly add to the overall scientific merit of this publication. 

2) The authors in the data availability section link a dataset on zenodo, which is fine. However, since the raw data of the primary experiment are shotgun metagenomics reads, those should be uploaded (after careful removal of all human DNA sequences) to a public repository (e.g. SRA) and the ID should be provided.

3) The authors should consider looking for this phage not only in the assembled phage genomes, but also testing other publicly available shotgun human virome datasets.

Minor points:

1) The PhASS name suggested for the new putative phage genus is questionable. a name which recalls some biological feature of the phages might be more appropriate

2) The authors state that they have looked for homologous sequences using BLAST, however it is not entirely clear on which database they performed their search. This should be clearly stated.

1) the word "conservative" is used wrongly in many instances to describe evolutionary conserved (not conservative) features of phages.

2) I recommend extensive revision of the English and adoption of a more formal language, avoiding expressions like "a lot of"

Author Response

Point-by point response to the comments of the Reviewer 1.

Comments and Suggestions for Authors

The authors report the identification of a novel phage from a single specimen. 

Major points:

1) a major limitation of this study is in the analysis of just one single sample. Analysis of a cohort of samples would greatly add to the overall scientific merit of this publication. 

- In fact, twenty virome shotgun libraries were constructed from fecal samples of healthy volunteers. After sequencing and de novo assembly, the unusual, but clear phage complete genome was identified only in one virome. We add this at the beginning of the results.

2) The authors in the data availability section link a dataset on zenodo, which is fine. However, since the raw data of the primary experiment are shotgun metagenomics reads, those should be uploaded (after careful removal of all human DNA sequences) to a public repository (e.g. SRA) and the ID should be provided.

- We have submitted the raw data to the BioSample database. BioSample accession: SAMN37731570. Temporary SubmissionID: SUB13890116. They send the release date as soon as processing is complete and we upload to the manuscript.

3) The authors should consider looking for this phage not only in the assembled phage genomes, but also testing other publicly available shotgun human virome datasets.

- In addition to the assembled phage genomes, the available human, animal, and environmental microbiomes were screened. However, only two similar genomes were found in human microbiomes (Lines 106-115) and a group of genomes with lower similarity then those were identified in human and animal microbiomes (L 197-212) (see Table 1 and Fig 3B). In addition, two phage genomes with low, but at least some similarity were found and used in VIPTree (Fig. 3). Unexpectedly, even fragments of genomes similar to phAss-1 were not revealed in the screened databases. This search was described in 2.2.

Minor points:

1) The PhASS name suggested for the new putative phage genus is questionable. a name which recalls some biological feature of the phages might be more appropriate

- We would prefer to keep this name. The fact is that we have no data on biological feature of this phage and its relatives. In addition, CrAss-phages exist (after cross assembly); phAss-1 phage was assembled from one sample.

2) The authors state that they have looked for homologous sequences using BLAST, however it is not entirely clear on which database they performed their search. This should be clearly stated.

- Two complete genomes with substantial similarity to phAss-1 were found in the metagenome data as a result of BLASTN search using nt_viruses and nr/nt databases (The last contains all GenBank, EMBL, DDBJ, and PDB sequences, excluding sequences from PAT, EST, STS, GSS, WGS, TSA and phase 0, 1 or 2 HTGS sequences). We note nt_viruses and nr/nt databases in the revised version.

Comments on the Quality of English Language

1) the word "conservative" is used wrongly in many instances to describe evolutionary conserved (not conservative) features of phages.

- Corrected

2) I recommend extensive revision of the English and adoption of a more formal language, avoiding expressions like "a lot of"

- Removed

Reviewer 2 Report

In their publication, Babkin et al present the assembly of a phage contig derived from a virome of human gut. They show that several open reading frames contain TAG stop codons. In addition, the genome encodes three suppressor t-RNAs which allow decoding of the ORFs to proteins of known function. This is followed by an extensive comparative analysis of viromes and the establishment of a novel group of phages within the proposed virus genus of Phassvirus. the analysis is very intersting since it unravels seeral viruses with TAG stop codons in ORFs as well as supressor t-RNA. However, also phage genomes with TAGs are found not encoding supressor t-RNAs as well as genomes free of TSGs in ORFs. Thus, they suggest to use for phage genome assembly various translation tables they provide. The study is done very carefull and appears comprehensive and interesting. A few minors points should be adressed.

1. English, see below.

2. Results: the results should start with what was done. To me it is unclear whether the contigs were derived from a data base or from own establishment of the gut virome of a healthy donor. the description on M&M comes too late or understanding.

3. Fig. 2: the two ribbon models should be shown in the same orientation and size. Why is in Bt he connection between N- and C-terminal part  missing.

4. Fig. 3: some of the panels are to small. It is not possible to read the letters. Same for Fig. 4 and 5.

5. Fig. 6: no open circles are found. small asterrixes appear.

6. page 12 second pragraph. This paragraph is completely ununderstandable. The SU10 capsid protein is introduced but suddenly the authors discuss Ig-lik domains. Please explain.

7: line 336: "phASS-1" should be deleted. it is probaly errenously found here.

8. Discussion. The authors assume that the evolutionary advantage of the TAG plus supressor t-RNA provides a mechanism of gene regulation. It would be intersting wha the authors feel about TAGs in ORFs and no supressor t-RNA in the genome.

By enlarge the English is very good. Only on a few locations it could be improved. This is especially needed for a few very long sentences which should be restructured to be easier understandable.

Author Response

Point-by point response to the comments of the Reviewer 2.

Comments and Suggestions for Authors

In their publication, Babkin et al present the assembly of a phage contig derived from a virome of human gut. They show that several open reading frames contain TAG stop codons. In addition, the genome encodes three suppressor t-RNAs which allow decoding of the ORFs to proteins of known function. This is followed by an extensive comparative analysis of viromes and the establishment of a novel group of phages within the proposed virus genus of Phassvirus. the analysis is very intersting since it unravels seeral viruses with TAG stop codons in ORFs as well as supressor t-RNA. However, also phage genomes with TAGs are found not encoding supressor t-RNAs as well as genomes free of TSGs in ORFs. Thus, they suggest to use for phage genome assembly various translation tables they provide. The study is done very carefull and appears comprehensive and interesting. A few minors points should be adressed.

  1. English, see below.
  2. Results: the results should start with what was done. To me it is unclear whether the contigs were derived from a data base or from own establishment of the gut virome of a healthy donor. the description on M&M comes too late or understanding.

- In fact, twenty virome shotgun libraries were constructed from fecal samples of healthy volunteers. After sequencing and de novo assembly, the unusual but clear phage complete genome was identified only in one virome. We add this at the beginning of the results.

  1. Fig. 2: the two ribbon models should be shown in the same orientation and size. Why is in Bt he connection between N- and C-terminal part  missing.

- We correct the figure and it is in the same orientation and size now. In “B”, the connection is missing as “B” represents two proteins (tail fiber protein and the protein containing two pectate lyase domains) that originated in phage phAss-1 (Fig 2B) instead of one protein in two relative phages (Fig. 2A). The fact is that there is one ORF encoding one protein (the receptor-binding tail fiber protein, 1070 aa) in the BK046881_ctckW2 and BK025033_ct6IQ4 genomes of the closely relative phages. However, this ORF is divided into two ORFs with the short intergenic spacer in the phAss-1 genome. That is why there are two proteins in phage phAss-1.

  1. Fig. 3: some of the panels are to small. It is not possible to read the letters. Same for Fig. 4 and 5.

- This is a standard problem. When final version is accepted, all panels are usually readable. Please, try to increase the scale of the figures. In addition, we increase the smallest panel in the Fig 3.

  1. Fig. 6: no open circles are found. small asterrixes appear.

- Indeed, “The investigated sequence of phAss-1 is marked with a black circle; corresponding sequences of relative phages from the proposed Phassvirus genus are indicated with empty circles; sequences from phage genomes that are listed in Table are marked with empty triangles. Nodes with 95% statistical significance are marked with asterisks calculated from 1000 ultrafast bootstrap (UFBOOT) replicates.”

  1. page 12 second pragraph. This paragraph is completely ununderstandable. The SU10 capsid protein is introduced but suddenly the authors discuss Ig-lik domains. Please explain.

- Indeed, this needs clarification.

Analysis with the use of protein structure homology-modeling SWISS-MODEL server showed the similarity of the above N-terminal sequences with that of Escherichia coli phage SU10 nozzile protein that contains several immunoglobulin-like (Ig-like) domains [21]. InterProScan and BLASTP analyses confirmed that Ig-like domain is located in the N-terminal part of the putative Hoc protein of phAss-1. …

7: line 336: "phASS-1" should be deleted. it is probaly errenously found here.

- Deleted

  1. Discussion. The authors assume that the evolutionary advantage of the TAG plus supressor t-RNA provides a mechanism of gene regulation. It would be intersting wha the authors feel about TAGs in ORFs and no supressor t-RNA in the genome.

- Possibly, these phages with their hosts fell into relatively new niche or the phages change their hosts (possibly to closely related ones) and TAG-suppression mechanism has become unnecessary. In this case, the presence of TAGs in ORFs without supressor t-RNA in the phage genome is just a "memory of a former life". However, this is an unsubstantiated hypothesis.

Comments on the Quality of English Language

By enlarge the English is very good. Only on a few locations it could be improved. This is especially needed for a few very long sentences which should be restructured to be easier understandable.

  • We try to find very long sentences and restructure them.

Reviewer 3 Report

Title: Genomes of a Novel Group of Phages that Use Alternative Genetic Code Found in Human Gut Viromes

The authors have reported an unusual phage genome (phAss-1, 135,976 bp), which they identified by de novo assembly of the human gut virome. Via genome analysis, they have observed the presence of the TAG stop codons in 41 ORFs, including characteristic phage ORFs, and three genes of suppressor tRNA.  Additionally, the authors have observed two phage genomes (BK046881_ctckW2 and BK025033_ct6IQ4) with substantial similarity to phAss-1 were extracted from the human gut metagenome data. The authors have shown that several of the monophyletic groups of divergent phage genomes, containing the proposed Phassvirus genus genomes, also have ORFs with suppressor TAG stop codons, and the need to use different translation tables when submitting phage genomes to GenBank.

The study has a sound methodology.

The analysis has been described well. 

The discussion is clear and balanced.

I have a few minor comments:

Line 165: Full stop after ‘despite low similarity’

Figure 2 Figure 8: Which software was used to create the ribbon structures? Please add to the Figure legend.

Author Response

Point-by point response to the comments of the Reviewer 3.

Title: Genomes of a Novel Group of Phages that Use Alternative Genetic Code Found in Human Gut Viromes

The authors have reported an unusual phage genome (phAss-1, 135,976 bp), which they identified by de novo assembly of the human gut virome. Via genome analysis, they have observed the presence of the TAG stop codons in 41 ORFs, including characteristic phage ORFs, and three genes of suppressor tRNA.  Additionally, the authors have observed two phage genomes (BK046881_ctckW2 and BK025033_ct6IQ4) with substantial similarity to phAss-1 were extracted from the human gut metagenome data. The authors have shown that several of the monophyletic groups of divergent phage genomes, containing the proposed Phassvirus genus genomes, also have ORFs with suppressor TAG stop codons, and the need to use different translation tables when submitting phage genomes to GenBank.

The study has a sound methodology.

The analysis has been described well. 

The discussion is clear and balanced.

I have a few minor comments:

Line 165: Full stop after ‘despite low similarity’

  • Corrected.

Figure 2 Figure 8: Which software was used to create the ribbon structures? Please add to the Figure legend.

  • 3D models were predicted using AlphaFold2 and rendered using UCSF Chimera molecular visualizer, version 1.17. This phrase is added to the figure legends.